# Antenna Protein Clustering In Vitro Unveiled by Fluorescence Correlation Spectroscopy

**DOI:** 10.3390/ijms22062969

**Published:** 2021-03-15

**Authors:** Aurélie Crepin, Edel Cunill-Semanat, Eliška Kuthanová Trsková, Erica Belgio, Radek Kaňa

**Affiliations:** 1Centre Algatech, Institute of Microbiology of the Czech Academy of Sciences, Opatovický Mlýn, 379 81 Třeboň, Czech Republic; semanat@alga.cz (E.C.-S.); kuthanova@alga.cz (E.K.T.); kana@alga.cz (R.K.); 2Faculty of Science, University of South Bohemia in České Budějovice, Branišovská 31a, 370 05 České Budějovice, Czech Republic

**Keywords:** photosynthesis, photoprotection, antenna proteins, non-photochemical quenching, fluorescence correlation spectroscopy, detergent critical micelle concentration, protein oligomerization

## Abstract

Antenna protein aggregation is one of the principal mechanisms considered effective in protecting phototrophs against high light damage. Commonly, it is induced, in vitro, by decreasing detergent concentration and pH of a solution of purified antennas; the resulting reduction in fluorescence emission is considered to be representative of non-photochemical quenching in vivo. However, little is known about the actual size and organization of antenna particles formed by this means, and hence the physiological relevance of this experimental approach is questionable. Here, a quasi-single molecule method, fluorescence correlation spectroscopy (FCS), was applied during in vitro quenching of LHCII trimers from higher plants for a parallel estimation of particle size, fluorescence, and antenna cluster homogeneity in a single measurement. FCS revealed that, below detergent critical micelle concentration, low pH promoted the formation of large protein oligomers of sizes up to micrometers, and therefore is apparently incompatible with thylakoid membranes. In contrast, LHCII clusters formed at high pH were smaller and homogenous, and yet still capable of efficient quenching. The results altogether set the physiological validity limits of in vitro quenching experiments. Our data also support the idea that the small, moderately quenching LHCII oligomers found at high pH could be relevant with respect to non-photochemical quenching in vivo.

## 1. Introduction

Photosynthesis is the process at the basis of almost all life on Earth; it allows conversion of sunlight energy into chemical energy. The very first step of the process, the harvesting of light, is performed by so-called “antenna proteins”, which bind hundreds of pigment molecules, mainly chlorophylls and carotenoids. The most well-known model for these pigment proteins is the major antenna of the photosystem II (PSII) of green organisms, LHCII. It is organized as trimers of Lhcb1-3 isoforms, each monomer counting 3 transmembrane helices and Times New Roman binding 14 chlorophyll molecules and 4 carotenoids [1,2].

These antennas are associated with photosystem complexes and play a crucial role in transferring the absorbed energy to the core. However, they are also involved in another essential function—the dissipation of excess light energy. In conditions where the electron transport chain gets saturated, photosynthetic organisms may face a potentially harmful over-absorption of light energy that cannot be converted into chemical energy. A number of regulatory mechanisms thus allow safe dissipation of excess energy; most of them are regrouped under the appellation of non-photochemical quenching (NPQ). The largest part of NPQ, the energy-dependent qE, consists of the pH-dependent de-excitation of chlorophyll-excited states through heat [3]. It is therefore assessed in an indirect way, on the basis of the measured decrease in fluorescence emission (quenching) [4]. It is called energetic quenching as it is triggered, in a matter of seconds to minutes, by luminal acidification concurrent with light illumination. Several different processes are suggested to be at the basis of qE, amongst which the widely spread xanthophyll cycle (the reversible de-epoxidation of violaxanthin in zeaxanthin) or specific proteins sensitive to low pH (PsbS in the green lineage, LHCSR/Lhcx in algae [5]). It has also been proposed that lumen acidification directly acts on antenna proteins, inducing their protonation and, in turn, aggregation [6,7,8,9,10]. The basis for this model is the observation that purified antenna proteins artificially aggregated in vitro present similar spectral properties to those seen in vivo when fluorescence quenching occurs [11,12]. Moreover, the extent of quenching within solubilized antennas seem to correlate with buffer acidification (see, for example, [13,14,15]). A typical feature often associated with energetic quenching is a red shift in LHCII fluorescence emission [16,17]. This specifically consists in an increase in fluorescence around 700 nm, contrasting with the usual single sharp 680 nm emission peak of solubilized trimers [11,18]. Its origin is still debated. It has been put in relation with a physiological alteration of chlorophyll energy levels within antennas [19]; however, other optical effects cannot be excluded. In artificial membrane systems, for example, this seems to be more pronounced in presence of aggregates [20,21].

The in vitro quenching method has been widely used in the past three decades to study energy dissipation in antenna proteins of various organisms, including plants [22,23,24,25] and algae [14,26,27]. Most of these studies use a similar protocol—detergent content is decreased either by sample dilution (see [14,24,25,27]) or by dialysis or titration with Bio-Beads, which remove almost all detergent from the solution, but without clear control [22,28,29,30]. At the same time, the protocol requires lowering the pH of the medium to enhance fluorescence quenching. Whilst it is generally thought that low (<6.5) pH is the trigger for qE activation, a direct measure of lumenal pH is not straightforward and there is still debate on the actual lower limit range [31,32,33]. Several authors claim that the pH of the lumen cannot go below pH 5.5 or even 6.0 [34,35]. However, most of in vitro quenching experiments are commonly performed between pH 5.0 and 3.5 (see, for instance, [25,36]), and little is known on the organization and size of the particles obtained at these pH conditions. The physiological relevance of the results obtained in vitro is therefore debatable for the in vivo situation in thylakoids.

Among all the methods that can be used to clarify these points, fluorescence correlation spectroscopy (FCS; described in Appendix A) is a promising candidate. As a confocal microscopy method, it can resolve temporal fluctuations in fluorescence intensity of particles diffusing through a very small volume (around 1 fL). Photophysical parameters can be accurately estimated from FCS data, obtaining valuable information about changes in the states of the sample (such as changes in size, which is interlinked with diffusion, see the Section 4). The raw FCS data is interpreted through an autocorrelation function (ACF) (Appendix A), which is subsequently further analyzed (Appendix A). Several fitting routines can be applied for the analysis, with the Levenberg–Marquardt multi-tau algorithm [37] being the most used to obtain average values (see the Section 4 and Appendix A). The results can also be fitted using the maximum entropy method (MEMFCS) [38], a multi-component data algorithm capable of distinguishing between several diffusing species [38,39], which allows breaking down the average diffusion times into several components, unveiling sample heterogeneity. Such investigation presents the advantage of being model-free, and thus rules out the uncertainty that comes from the manual determination of the number of components.

Compared to other methods providing similar information, FCS presents the advantage of studying the molecules directly in solution, and allows direct measurement of the fluorescence properties of the particles on top of their size. Although it has been very minimally used in photosynthesis to this day, it has already proven to be applicable to study the diffusion properties and size of photosynthetic antennas [40,41,42] and other thylakoid membrane proteins [43].

Here, we took a step further and used FCS to shed new light on the in vitro clustering of plant LHCII antennas, in association with classical, bulk quenching experiments in detergent solution. This approach allowed us to estimate fluorescence quenching and LHCII particle size at the same time. The results, on one hand, disclosed clear limits of in vitro quenching experiments, showing that certain experimental conditions induce the formation of large oligomers up to aggregates unlikely to be physiologically sound. On the other hand, they supported the idea that a slow, mild quenching, such as the one found in small oligomers, may better represent non-photochemical quenching in vivo.

## 2. Results

### 2.1. Effect of pH and Detergent on LHCII Fluorescence Quenching: Bulk Measurements

In vitro fluorescence quenching is a commonly used method to induce fluorescence quenching of isolated antenna proteins by means of detergent dilution often in association to pH drop [14,25,28,29,30,44,45]. When LHCII antennas were injected in a buffer of pH 7.3 and 200 µM (final concentration) of n-dodecyl-β-D-maltoside (hereafter DDM), their fluorescence signal decreased by ≈15% (quenching K_D_ = 0.1) compared to the initial signal due to light-induced effects [44,46] (Figure 1A, Table 1). At pH 7.3, 100 µM DDM, fluorescence quenching was about 40% (K_D_ = 0.7), due to the halved detergent concentration. Finally, at pH 7.3, 25 µM DDM, quenching was about 60% (K_D_ = 1.4), for an overall almost linear trend between detergent dilution and fluorescence quenching (Figure 1C). When the same experiment was performed at pH 5.0 (Figure 1B), the extent of quenching was not proportional to the decrease in detergent concentration. From 200 to 100 µM DDM, a threefold increase in quenching was observed (corresponding to quenching K_Ds_ of 0.2 and 2, respectively, see Figure 1C and Table 1). Finally, at 25 µM DDM, the fraction quenched was 75% corresponding to a K_D_ of 3 (Table 1). The most evident effect of pH, however, concerned quenching rate constants (Q_R_, see also the Section 4). Whilst at pH 7.3, Q_R_s were constant and the trend was flat (Figure 1D), their value increased exponentially at pH 5.0, confirming a synergic action of detergent depletion and acidification [25,47]. In order to assess the physical properties of particles formed during the oligomerization, we applied fluorescence correlation spectroscopy (FCS) to samples prepared using the same “in vitro quenching” protocol.

### 2.2. FCS Unveiled Sizes of LHCII Clusters

FCS measuring protocol was applied to determine LHCII antenna particles sizes on the basis of their diffusional properties in solution (Appendix A). Samples were prepared in the same way as for bulk measurements (Figure 1). First, the measured autocorrelation functions (ACFs) were fitted by Levenberg–Marquardt method (see the Section 4), using one diffusional component for each type of sample (Figure 2 and Table 2, as well as Appendix A for raw curves and weighted residuals of the fittings, respectively). At 200 µM DDM, samples incubated either at pH 7.3 or at pH 5.0 displayed similar diffusion times of 1.18 and 1.10 ms, respectively (Table 2). The values corresponded to apparent radius of 9.2 nm for pH 7.3 and 7.9 nm for pH 5.0 (see Table 2). These values were compatible with behavior of a trimeric LHCII in detergent [41,48], confirming lack of a significant pH effect on LHCII oligomerization state above detergent critical micellar concentration (CMC). At pH 7.3, a low order oligomerization started to be visible below 100 µM DDM, with average particle radii ranging between 13 and 15 nm at 100 and 25 µM, respectively (Table 2). Interestingly, at this pH, a further decrease in detergent concentration (down to 25 μM DDM) did not yield any detectable increase in LHCII clustering (Figure 2). On the contrary, at pH 5.0 and 100 µM DDM, a twofold increase in particle radius was found, corresponding to about 30 nm. This result indicates that low pH deeply affects LHCII antenna protein cluster sizes. Even larger protein oligomers (of apparent radius over 65 nm) seemed characteristic of the pH 5 and 25 μM DDM condition (see Figure 2, Table 2). However, their presence is based on calculations that are distorted by the quality of the fittings (see Appendix A), and must therefore be taken with caution (see the Section 3).

### 2.3. Low pH Induced LHCII Large Particle Clusters

Besides particle size, the Levenberg–Marquardt method of ACFs fitting allowed for simultaneous characterization of the fluorescence properties of LHCII, in terms of total photon count rate (see Table 2), and average number of LHCII clusters per focal volume (N, Table 2). Photon count rate data were in good agreement with bulk data (see Figure 1 and Table 2). Concerning the number of particles, above detergent CMC (200 μM DDM), no major differences were observed between low and high pH (Table 2). A marked difference was instead evident below the detergent CMC. At high pH, the apparent number of particles in the confocal volume decreased between 200 μM and 100 μM DDM by ≈30% (from 35.0 to 23.5) and then again by ≈30% at 25 μM DDM (see Table 2). It confirmed a “mild”, seemingly controlled detergent-induced particle clustering at high pH. The drop in particles per volume was instead very pronounced at pH 5.0; it decreased from 36.3 at 200 μM DDM to 4.7 at 100 μM DDM, pointing at a more abrupt clustering process induced by low pH.

### 2.4. MEMFCS Analysis Highlighted LHCII Cluster Size Heterogeneity at Low pH

In order to uncover potential heterogeneity in particle size, we applied maximum entropy method fitting of FCS data (MEMFCS) (see Appendix A and the Section 4). This is a well-established, model-free method typically employed for heterogeneous samples [38]. The method also determines the number of components in the sample when their respective brightness is unknown [49,50,51]. The MEMFCS analysis showed that one main component was sufficient to fit satisfactorily samples prepared at pH 7.3 (see Figure 3A, and residuals in Appendix A). The diffusion times were comparable to those obtained by classical Levenberg–Marquardt analysis (Table 2). These samples were therefore represented by quasi-homogeneous populations of radii ranging from 8 to 30 nm. Instead, at 25 μM DDM, pH 7.3, beside the 30 nm component, a seemingly novel component appeared, characterized by an apparently short diffusion time of around 0.3 ms (Appendix A). This was interpreted as due to photophysical dynamics of the protein (see the Section 3). At pH 5.0, the clustering process appeared to be different (Figure 3B). Whilst one main trimeric population was present in the sample at 200 μM, decreasing detergent below the CMC resulted in the appearance of heterogeneous populations of LHCII, some trimeric (with radii around 8 nm at 100 μM DDM), and others made of larger oligomers or small aggregates of apparent radii up to 60 to 80 nanometers. A similar behavior was apparent at 25 μM DDM; however, in this case, the mathematical model failed to properly describe this condition (see Appendix A and the Section 3).

### 2.5. Aggregates Out of Physiological Range Induced by In Vitro Fluorescence quenching

Within the photosynthetic literature, it is not uncommon to use very high protein to detergent ratio and low pH to induce and investigate non-photochemical quenching [24,25,28,29,30,52]. As these types of samples were found problematic to study by FCS (see 25 μM DDM, pH 5.0 condition in Appendix A and Table 2), we applied classical confocal analysis instead to estimate sizes and fluorescence of LHCII clusters. As shown in Figure 4A,B (original confocal picture and detected particles, respectively), acidic samples prepared far below CMC were characterized by LHCII clusters of cross-section areas between 1 and 10 µm^2^. The particles were variegated in size (Figure 4A,B) and fluorescence intensity per particle (Figure 4C). The size distribution calculated on the basis of the polydispersity index (PDI) (also called heterogeneity index, see the Section 4 and [53]) had a value of 1.1, indicative of highly disperse, multi-sized populations of LHCII trimers. When the fluorescence per particle was plotted as a function of the cross-section surface, the graph showed an interesting trend. Below 3 µm^2^, it increased linearly with particle size (*R*^2^ of straight-line fit = 0.8, see Figure 4C); above this value, any further increase in size did not correspond to an increase in fluorescence intensity. Particles were also characterized by a marked fluorescence heterogeneity; for cross-sections between 2.4 and 2.8 µm^2^, for example, fluorescence intensities of between 9071 and 18,072 (arbitrary fluorescence units) were found, with an up to 100% increase in fluorescence for a 17% area increase.

These results altogether seem to indicate that the harshest conditions often employed during in vitro quenching induce not just fluorescence quenching, but also a seemingly random LHCII particle organization, probably far from what is considered a finely tuned physiological process of non-photochemical quenching (see the Section 3).

### 2.6. Assessment of the Red-Shifted Fluorescence in the Samples

NPQ is sometimes put in direct relation with an increased emission of the redmost part of the spectrum (see the Section 1 and, for instance, [16,17]). We assessed the extent of red shift in LHCII by measuring low-temperature steady-state fluorescence of particles formed in the same conditions used for in vitro quenching (Figure 1). Our results showed that, at pH 7.3, only a very weak red shift was visible in samples incubated below detergent CMC (Figure 5A). This was visible from the ratio of the fluorescence emission peaks at 700 and 680 nm (F700/F680, Figure 5C). F700/F680 at pH 7.3 increased from 0.06 to 0.09 between the higher (200 µM DDM) and the lower (25 µM DDM) detergent concentrations, respectively. In contrast, at pH 5.0, the F700/F680 ratio was more than doubled (0.07 to 0.17) after lowering DDM from 200 μM to 25 μM DDM (Figure 5B,C), although it stayed very low compared to previously reported values [18,54]. Only by significantly increasing the protein to detergent ratio were we able to detect a clear, red-shifted emission band in our samples (up to a F700/F680 ratio of 0.48 at pH 5.0 and 100 µM DDM, see Appendix A). Interestingly, confocal microscopy provided an indication that large LHCII aggregates display red-shifted fluorescence (Appendix A). Intra-aggregate heterogeneity was observed in the emission—most aggregates mainly displayed red fluorescence (displayed here in blue), while some of them presented areas of far-red fluorescence (pink). These results altogether suggest that red shifts, rather than with non-photochemical quenching, relate with protein aggregation.

## 3. Discussion

Antenna aggregation has long been considered a major molecular mechanism of energy dissipation in photosynthetic organisms, called non-photochemical quenching (NPQ). However, most protocols used to study NPQ in vitro involve detergent removal and pH drop, yielding particles of unknown size and organization (see for instance [14,25,27]). This casts doubts on the relevance with respect to the in vivo non-photochemical quenching mechanism. To address this point, we used FCS as a well-established method to assess the oligomerization state of membrane protein complexes in a surfactant solution. FCS allowed for a simultaneous estimation of the oligomerization state of LHCII trimers and some of their fluorescence properties upon detergent depletion and/or acidification. We were thus able to validate the direct effect of both these parameters on LHCII clustering (namely, oligomerization and/or uncontrolled aggregation), as well as heterogeneity in the particles formed. The fluorescence quenching observed at this quasi-single molecule level was also comparable to the one observed in more traditional, bulk measurements.

It is first necessary to make a parenthesis concerning the correct interpretation of the FCS results and goodness of the fittings. One concern relates with the overlap of times purely due to particle diffusion with those due to photophysics in LHCII trimers. Indeed, most photophysical processes occurring in LHCII trimers fall out of range of the typical diffusion part of the autocorrelation curve. However, in the most extreme quenching conditions, photophysical effects tend to overlap with diffusion times [55,56]. For samples prepared at 25 µM DDM at pH 5.0, for instance, the extent of overlap of photophysics processes impeded proper fitting (Appendix A). One possible explanation for this is that at low DDM concentration, the LHCII complexes are more exposed to the environment, and a drop in pH induces a modification in the energetic landscape of the proteins, leading to new conformational transitions [57]. This would in turn cause changes in the energetic coupling between pigments [58], bringing new sceneries where LHCII presents different photophysical behavior, with consequent overestimation of the oligomers’ size (see, for instance, [59,60]).

Another limit of FCS interpretation concerns samples characterized by particle and pigment heterogeneity (see Figure 3 and Appendix A). The FCS models currently available are designed for systems of proteins containing only a few fluorescent chromophores [61,62]. A single LHCII trimer counts 42 chlorophylls and 12 carotenoids [63]. Upon LHCII oligomerization, two opposing processes occur simultaneously; on one hand, fluorescence increases with the size of the LHCII cluster due to an increased number of chlorophylls, on the other, protein oligomerization leads to fluorescence quenching [64]. As a consequence, the fluorescence yield of LHCII clusters is not fixed and the disentanglement between brightness of different components is not currently feasible. This is the reason why it was not possible to extract information concerning the concentration of different antenna clusters for heterogeneous samples. Nevertheless, by means of MEMFCS, we were able to decipher which conditions yielded sample heterogeneity at low and high pH, respectively (compare Figure 3A,B).

Despite the limits mentioned above, our FCS data brought few clear results. Fitting of FCS traces at 200 µM showed that LHCII does not cluster in detergent concentrations above CMC (Figure 2B and Figure 3). This is in accordance with previous studies [25]. Further, there was no pH effect above CMC, likely due to a large belt of detergent in micelles, which protects antenna complexes from the action of pH. Lack of pH effect was observed for all fluorescence properties analyzed here: kinetics, red shift, bulk, and single-molecule quenching. This point reinforces the idea that the changes in fluorescence observed during in vitro LHCII fluorescence quenching reflect a change in the structural organization in the protein environment.

Further, FCS revealed a specific pH effect on LHCII clustering. Whilst at high pH, even below detergent CMC, fluorescence quenching was always characterized by a slow rate, low pH instead accelerated the quenching process (Figure 1). On the basis of FCS data, we can suggest that this is the reflection of a different particle organization in the two conditions. In fact, an ordered particle clustering process, yielding small and homogeneous oligomers, was found at high pH (Figure 2B and Figure 3A). At low pH, instead, results evidenced the formation of larger (>30 nm) and heterogeneous particles (Figure 2B and Figure 3B). This seemingly uncontrolled oligomerization typical of low pH conditions can, in principle, explain the different, exponential fluorescence quenching kinetics found typical of in bulk pH 5.0 measurements (Figure 1D, see also [25] and references therein). It is interesting to point out that, according to our results, the quenching extent did not correlate with particle size. As summarized in Appendix A, in bulk fluorescence measurements showed that samples incubated at 25 µM DDM, pH 7.3, displayed fluorescence quenching amounts very different from the LHCII incubated at 100 µM (FQ ≈ 62% and 44%, respectively), even though the particle size was similar. Their quenching was instead closer to the sample incubated at 100 µM DDM and pH 5.0 (FQ ≈ 71%). On the basis of this, we tend to suggest that low pH is more effective on protein–protein clustering and quenching kinetics and less on fluorescence quenching yields.

The apparent diameter found for single trimers in solution by FCS (Table 2, just under 20 nm) is in line with the previous light scattering study [48]. It is larger than the diameter of LHCII trimers alone as determined by electron microscopy (7 to 8 nm, see, for instance, [65,66]). This is because FCS does not estimate the actual diameter of the molecules, but their hydrodynamic radius. It thus also takes into account (a) the ring of natural lipids co-purified with the trimers, which can be quite large in our mildly solubilized proteins; (b) the ring of detergent or pseudo-micelles, which depends on detergent concentration and is known to affect the measured diameter of protein complexes [67]; (c) the hydrodynamic volume moving with the particle. Indeed, our samples presented shorter diffusion times than previously observed in higher detergent concentrations [42]. However, it is difficult to estimate how these parameters, especially the detergent and hydrodynamic rings, are affected during the oligomerization. Combined with the fact that the actual structure of the oligomers is unknown, it is difficult to estimate the number of LHCII trimers involved in an oligomer from its calculated diameter only. The evolution of the number of particles, also estimated by FCS measurements, however, gives us hints on this point. For instance, the number of particles detected throughout the experiment changed from around 35 (at pH 7.3 and above detergent CMC) to an apparent 5 particles (at pH 5.0 and around CMC (Table 2)). This suggests the pH-induced formation of assorted low order oligomers, accounting for sample heterogeneity. Interestingly, particles of seven associated LHCII trimers have been observed by electron microscopy in partially solubilized plant membranes pre-washed in a pH 6.5 buffer [68], meaning that LHCII clusters similar to the ones found here can indeed occur in vivo. In any case, our results show that these rather small LHCII oligomers would be sufficient to induce fluorescence quenching, and thus play a physiological role in NPQ, in accordance with the current models [7,69].

It is interesting to analyze our results in terms of extents of physiological quenching. In vivo studies of qE showed that K_D_ values between 1 and 1.5 are typical of wild type chloroplasts and leaves [13,70]. These values are well compatible with those obtained here in small oligomers at pH 7.3 at both 100 and 25 μM DDM (see Table 1). On the other hand, greater K_D_s, typical of in vitro quenching experiments and close to those found here at low pH at 25 μM DDM (K_D_ ≈ 3), seem uncommon in vivo. They are usually obtained by the artificial addition of proton enhancers to the chloroplast suspension (see, for instance, [13]) or rather characteristics of mutants (L17) and plants treated with lincomycin [71,72]. Whilst we cannot exclude that in some specific conditions even large oligomers may be physiologically relevant (see below), still our results cast doubts on the use of buffer acidification below pH 5.5 when combined with detergent depletion, to represent in vivo photoprotection. Indeed, typical values of lumen pH in excess light were suggested to lie in the range between pH 6.5 and 5.8, whilst lower values lead to photoinhibition ([34]; also reviewed in [15,73]). However, a significant part of in vitro experiments of LHCII quenching led to this day involve pH values below 5 [25,52] and extremely low detergent concentrations (for example, [24]) or use of Bio-Beads to remove almost all detergent from the solution [28,29,30].

It is, however, important to distinguish between the effects of detergent depletion and buffer acidification. Whilst at pH 7.3, even the lowest detergent concentration tested (25 μM DDM) did not impact on sample homogeneity, yielding rather uniform small oligomers, at pH 5.0, lowering detergent to 100 μM was sufficient to induce sample heterogeneity (Figure 3B). This evidence goes against a previous “dogma”, considering detergent depletion-induced quenching and low pH-induced quenching as part of a continuum, arising from the same mechanism, where acidification simply accelerates the rate at which maximum quenching is reached [10,47]. FCS data instead showed that pH and detergent have a clear different impact on protein clustering (Figure 2, Table 2). Presence of aggregates of size up to several micrometers was noted only at low pH (Figure 4). Importantly, some of these aggregates were even larger than the average grana size (200–600 nm [74,75,76]). These micrometric aggregates are thus unlikely to be physiologically relevant. Interestingly, however, “supergrana” of micrometric size were found in plants treated by lincomycin [77]. At this condition, the cores of the photosystems are drastically depleted, leading to membranes highly enriched in aggregated LHCII [71]. We can thus speculate that this enlargement of the grana with lincomycin may be caused by the organization of LHCII into these large aggregates, as other proteins (like photosystems) do not impede their formation.

Notably, thylakoids from lincomycin-treated [71,72] as well as other highly stressed plants [16,17] displayed a strong red-shifted fluorescence, as was often observed in in vitro quenching [18,54] or in model membrane systems [20,21]. Our results indicate that this shift is strongly reduced in vitro in conditions yielding small LHCII particles. Indeed, the low F700/F680 ratios that we obtained in these conditions are in agreement with the ones obtained in liposomes by Tietz and coworkers [78], where the protein amount was limited to on average six LHCII trimers per liposome. This similarity seems to support the fact that the extent of the red-shift is rising proportionally with the size of the aggregates [21]. Confocal microscopy with two detectors (Appendix A) revealed heterogeneity within the aggregates, with patches of red-shifted fluorescence emission in the large clusters. Such intra-particle heterogeneity is in line with previous measurements in fluorescence lifetime microscopy [56]. It could also fit models predicting the presence of LHCII trimers or monomers playing the part of quenchers in aggregates (see, for instance, [24,79]), especially as its apparition has been put in relation with chlorophyll–chlorophyll charge transfer quenching states [45,79]. Unfortunately, the low resolution of the technique and the lack of information on the organization of the aggregates or number of LHCII trimers involved did not allow determining the physiological relevance of the phenomenon. The results indicated, on one hand, that antenna oligomerization does not induce significant red shifts, even though quenching is present. On the other hand, they suggest that red shifts relate with protein aggregation—a process of questionable physiological relevance—rather than with quenching. Nevertheless, we cannot exclude that the red-shifted emission observed in some in vivo studies (for instance, [19]) could have a different origin and result from of a highly packed natural environment or related to some specific lipid effect.

Our main results showed that LHCII oligomerization is increased by low pH (see Figure 2 and Table 2). It is worth mentioning that the formation of large LHCII aggregates in vivo or in artificial membranes was found to be pH-independent in some other studies [16,20,21]. This clustering seems to occur in the absence of other proteins in the membrane, or, in vivo, when the photosystem cores have been largely depleted (for instance under lincomycin treatment [71]). Low pH is therefore not the only parameter that promotes aggregation. We suggest that fast and extensive oligomerization involves a combination of two mechanisms. The first is triggered by low pH, and the second one represents some other pH-independent oligomerization processes that might come at play in extreme stress situations [16,17]. Further studies are necessary to untangle these potentially separate components.

## 4. Materials and Methods

### 4.1. LHCII Purification

Thylakoid membranes were purified from store-bought spinach leaves. Dark-adapted leaves were rinsed, and thylakoids extracted as described previously [65] with the following modifications: B1 solution contained 0.4 M sorbitol, 2 mM MgCl_2_, 20 mM Tricine KOH (pH 7.8), 0.5% Bovine Serum Albumin, and 1 mM aminocaproic acid; the leaves grinded in this solution were filtered through muslin and cotton.

For complex purification, these thylakoids were washed in a solution of 4-(2-hydroxyethyl)-1-piperazineethanesulfonic acid (HEPES) KOH 10 mM (pH 7.3) and resuspended to a concentration of 1 mg of chlorophyll (Chl)/mL in HEPES KOH 10 mM (pH 7.3). They were then solubilized in 9.8 mM (0.5%) n-dodecyl-β-D-maltoside (hereafter DDM) to a final concentration of 0.5 mg Chl/mL. After centrifugation at 12,500× *g*, 4 °C for 10 min, the supernatant was deposited on sucrose gradients prepared by freeze-thawing of a solution of 0.6 M sucrose, HEPES KOH 10 mM (pH 7.3), and 98 μM (0.005%) DDM. The gradients were ultracentrifuged in a SW40 rotor (Beckman Coulter, Brea, CA, USA) at 40,000 revolutions per minute (RPM), 4 °C, for 20 h.

### 4.2. Bulk Fluorescence Kinetic Measurements

Isolated LHCII (OD_676_ = 1 cm^−1^), solubilized in 200 µM DDM (final concentration), were diluted 20 times, with constant stirring, at room temperature in a 10 mM HEPES or 2-(N-morpholino)ethanesulfonic acid (MES) buffer (pH 7.3 or 5.0). DDM concentration was either 200 µM (0.01%), 100 µM (0.0051%), or 25 µM (0.0013%), representing final protein to detergent ratios of 1:2520, 1:1260, and 1:315, respectively. Chlorophyll fluorescence was continuously monitored using a FL 3000 fluorometer (PSI, Brno, Czech Republic) under orange actinic light (635 nm, 200 µmol photons m^−2^ × s^−1^). Fluorescence fraction quenched (FQ, in percentage), quenching K_D_, and rate constants (Q_R_) were determined as previously described (see, for instance, [80,81]). Briefly, quenched fractions were calculated at each detergent concentration and pH as (F_0_−F_500_)/F_0_, with F_500_ the fluorescence intensity at 500 s (corresponding to steady state) and F_0_ the unquenched fluorescence at the beginning of the protocol. Quenching K_DS_ were calculated as (F_0_−F_500_)/F_500_. Quenching rate constants (Q_R_) were obtained by fitting each trace with a three-parameter hyperbolic decay function: y = (y_0_ + ab)/(b + x), where 1/b represents the Q_R_ of the process.

### 4.3. Fluorescence Correlation Spectroscopy

FCS measurements were performed with an inverted Zeiss LSM 880 laser scanning confocal microscope (Carl Zeiss Microscopy GmbH, Oberkochen, Germany) equipped with a C-Apochromat 40×/1.2 NA water immersion objective. The LHCII surfactant solution was placed in an 8-well chambered coverslip (Ibidi GmbH, Gräfelfing Germany), and chlorophyll auto-fluorescence was excited by a HeNe 633 nm wavelength laser (intensity approx. 1.0 µW) and detected at 642–695 nm by photon-counting GaAsP detector. The pinhole was adjusted to 70 μm aperture. The raw signals of chlorophyll a fluorescence fluctuations were detected during 100 s in total (10 s × 10 times with almost continuous acquisition) for each measurement. The autocorrelation function was processed through a digital signal correlator card. For each condition under investigation (200 μM, 100 μM, or 25 μM of DDM, corresponding in these experiments to protein to detergent molar ratios of around 1:8400, 1:4200, and 1:1050, respectively; 10 mM HEPES (pH 7.3) or MES (pH 5.0)), we made 3 independent repetitions. All measurements were performed at room temperature.

### 4.4. FCS Data Analysis—Fitting of Autocorrelation Functions

At first, the autocorrelation functions (ACFs) were fitted by the conventional Levenberg–Marquardt multi-tau algorithm [37] by Zeiss Zen Black software to obtain characteristic diffusion times. To avoid artifacts due to uncorrelated signal belonging to photophysical components [82], we performed the fitting routine starting from 10 μs in the ACF. The ACF traces with significantly larger characteristic time were removed from the analysis to avoid overestimation of the molecular sizes. The fluorescence traces were fitted using a three-dimensional free diffusion model (Equation (1)) with 1 or 2 diffusional components.
(1)$$G\left(\tau \right)=\frac{1}{N}\left(\frac{1}{1+\left(\frac{\tau }{\tau_{D}}\right)}\right){\left(\frac{1}{1+{\left(\frac{r}{l}\right)}^{2}\left(\frac{\tau }{\tau_{D}}\right)}\right)}^{\frac{1}{2}}$$

### 4.5. FCS Data Analysis—Characterization of the Oligomers by the Maximum Entropy Method Algorithm (MEMFCS)

The characterization of the LHCII particles by MEMFCS was carried out by employing the plugin *fcs_maxent* in the open-source data evaluation package QuickFit 3.0 [83]. We used a physical model of 3D normal diffusion (Equation (2)).
(2)$$G\left(\tau \right)=\overset{n}{\underset{i=1}{\sum }}\alpha_{i}\left(\frac{1}{1+\left(\frac{\tau }{\tau_{Di}}\right)}\right){\left(\frac{1}{1+{\left(\frac{r}{l}\right)}^{2}\left(\frac{\tau }{\tau_{Di}}\right)}\right)}^{\frac{1}{2}}$$
where *α_i_* is the amplitude of *i*th species, *r*/*l* is the ratio between radial and longitudinal dimensions of the confocal volume (known as structural parameter), and *τ_Di_* is the diffusion time of the particles in solution. In the classical maximum entropy theory, *α* represents a regularization parameter determined by successive iterations. Although proportional to particle brightness (see [38] for details), it does not require the prior knowledge of the number of components nor their respective brightness. The optimum maximum entropy solutions yield the distribution of the *α_i_* values that account for the best fitting when the entropy is maximized. The axial ratio was set to 6 and other local parameters such as triplet and triplet population were settled up with the values obtained from conventional FCS fitting routine with 1 additional triplet component in this occasion.

The goodness of the fitting was evaluated on the basis of the calculated weighted residual [38].

### 4.6. FCS Data Analysis—Estimation of the Size of LHCII Particles

The size of the LHCII particles were calculated from ACF fitting using the diffusion coefficient (*D*) starting from the diffusion time (*τ_D_*) following the equation (Equation (3)).
(3)$$\tau_{D}=\frac{\omega_{x}^{2}}{4D}$$

Here, *ω_x_* is the length of the confocal volume in the *x*-axis, determined using a standard Atto 647-carboxylic acid dye (MW: 746 g/moL, concentration: 30 nM, excitation: 633 nm, detection 642–695 nm) with an assumed diffusion coefficient of 4.1 × 10^−6^ cm^2^ × s^−1^ at 25 °C [84] and *ω_x_* ≈ 340 nm.

The Stoke–Einstein equation (Equation (4)) was used to calculate the hydrodynamic radius *R_H_* (with temperature *T* = 298.15 K; water dynamic viscosity, *η* = 8.91 × 10^−4^ Ns/m^2^; *k* is the Boltzmann constant).
(4)$$D=\frac{kT}{6\pi \eta R_{H}}$$

### 4.7. Standard Confocal Microscopy Imaging of LHCII Particles

µ-Slide 8-Well Chamber Slides were prepared for imaging by incubation at room temperature with 100 μL/well of 0.1% of poly-L-lysine (PLL) solution. After 5 min of incubation, the remaining solution was discarded, and the wells were washed 3 times with water. Then, the PLL-coated wells were incubated with a solution containing 10 nM of LHCII trimers at pH 5.0 and 6 μM of DDM, respectively, for 20 min and the bottom of the well was further imaged.

Confocal images of the immobilized protein aggregates were acquired by a laser scanning confocal microscope (Zeiss LSM 880; Carl Zeiss Microscopy GmbH, Oberkochen, Germany) equipped with a Plan-Apochromatic 63×/1.4 Oil DIC M27 objective (Carl Zeiss Microscopy GmbH, Oberkochen, Germany). The LHCII auto-fluorescence signal was collected by a GaAsP photomultiplier in 16-bit mode. The images sized at 103.81 × 103.81 μm were taken with the following parameters: zoom: 1.3; pinhole: 63 μm; pixel dwell time: 2.06 μs; frame 512 × 512 pixels; dichroic mirror: MBS 488. The auto-fluorescence of LHCII was excited with a 488 nm laser and detected in 2 channels, ChS1 624–695 nm and Ch2 696–758 nm. The acquired images were segmented and analyzed by the FIJI [85] version of ImageJ as 8-bit RGB pictures. The images were first converted from 16 to 8 bit and manually prepared (threshold processing) to eliminate the background. They were then analyzed by means of Analyze particle plugin, and the 3D surface plot plugin was used for presenting the particle density in the images. Data presented in Figure 4C were fitted by polynomial (linear) and sigmoidal (3 Hill parameter) functions using Sigmaplot Regression Wizard tool.

The polydispersity index (PDI), or heterogeneity index, was calculated as the square of the area standard deviation divided by the square of the area mean. PDI smaller than 0.05 indicate a highly homogeneous population; PDI over 0.7 point to a very broad particle size distribution [53].

### 4.8. Low-Temperature Fluorescence

Low-temperature steady-state fluorescence was measured on LHCII incubated 20 mM HEPES or MES buffer and the same protein and detergent concentrations as for the fluorescence kinetic measurements (see above), at the pH of interest, with an excitation at 475 nm and emission measured between 600 and 800 nm. Resulting curves were normalized on the maximum.

## 5. Conclusions

Our FCS results show that, below detergent CMC, low pH enhanced protein clustering. Far below CMC (25 μM or less), it induced formation of a highly heterogeneous mix of large, almost micrometer-sized LHCII particles. These experimental conditions, often employed during in vitro experiments with photosynthetic antennas, seem to induce rather uncontrolled aggregation. The consequent effects of fluorescence quenching are therefore probably not relevant to the finely tuned in vivo processes connected with photoprotection. However, milder detergent conditions at neutral pH instead promoted oligomerization of the LHCII in small complexes of under 10 trimers, closer to physiological conditions [68]. Our bulk measurements proved that these small particles are sufficient to induce fluorescence quenching, albeit slower than observed in extreme conditions (Figure 1). However, is strong and fast quenching—such as voluntarily pushed in most in vitro aggregation studies—really necessary for in vivo photoprotection? The recently proposed model of “economic photoprotection” [86] already suggested that slow quenchers offer more flexibility for light acclimation without competing with photochemistry, ensuring optimized photosynthesis at all times. The moderately quenching LHCII oligomers observed in the present study at high pH, would fit well with this model.

## Figures and Tables

**Figure 1 ijms-22-02969-f001:**
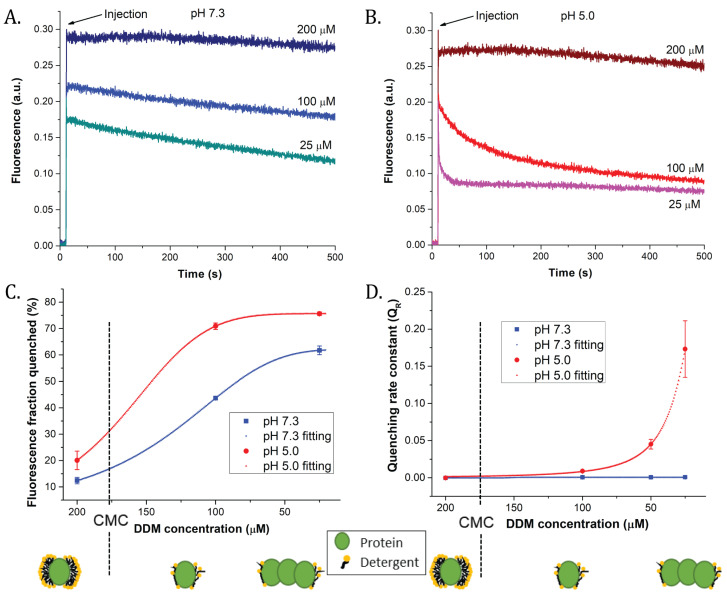
Fluorescence quenching analysis of LHCII trimers during bulk measurement in vitro. Panels (**A**,**B**) represent time course of fluorescence quenching of LHCII trimers (in HEPES 10 mM (pH 7.3) and 200 μM n-dodecyl-β-D-maltoside (DDM)) injected in a buffer of pH 7.3 (**A**) or 5.0 (**B**) at different detergent concentrations. (**C**) Titration curves of quenched fraction of fluorescence calculated from panels (**A**,**B**) (see the Section 4 for calculation). (**D**) Rate constants of fluorescence quenching, calculated by fitting time courses of fluorescence (**A**,**B**) with a sigmoidal Hill equation y = [axb]/[cb + xb] (see, for instance, [10,14]). Circles/squares = data average; dashed lines = data fittings; error bars: SD. Data are average of three measurements. The proposed organization of the detergent around the LHCII single trimer above and below critical micellar concentration (CMC) is represented under the graphs.

**Figure 2 ijms-22-02969-f002:**
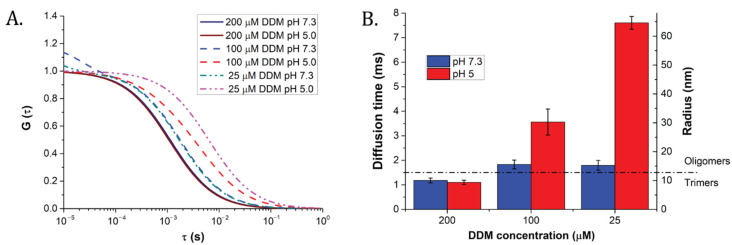
Averaged fittings of the autocorrelation functions and calculated diffusion times of LHCII trimers obtained by conventional (Levenberg–Marquardt) method. The full list of parameters obtained is presented in the Table 2. See Appendix A for raw curves and weighted residuals of the fittings, respectively. (**A**) Averaged autocorrelation curves. A shift to longer diffusion times indicates an increased size of LHCII clusters. (**B**) Diffusion times and particle radii estimated from them (see the Section 4). The diffusion time threshold for a trimeric state of LHCII is marked with a dashed line. Results are the average of three independent measurements. Error bars: SD.

**Figure 3 ijms-22-02969-f003:**
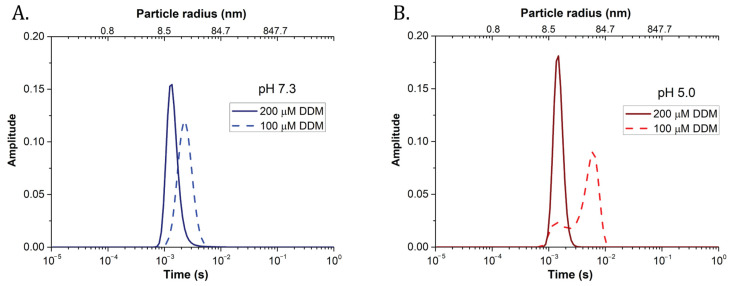
Maximum entropy FCS fitting method unveiled sample heterogeneity. Typical results obtained fitting FCS autocorrelation curves (see the Section 4) for samples prepared at 200 or 100 µM DDM, pH 7.3 (**A**), or pH 5.0 (**B**). The method allows for separating different diffusion components, highlighting possible size heterogeneity in LHCII clusters. The area below each trace was normalized to 1.

**Figure 4 ijms-22-02969-f004:**
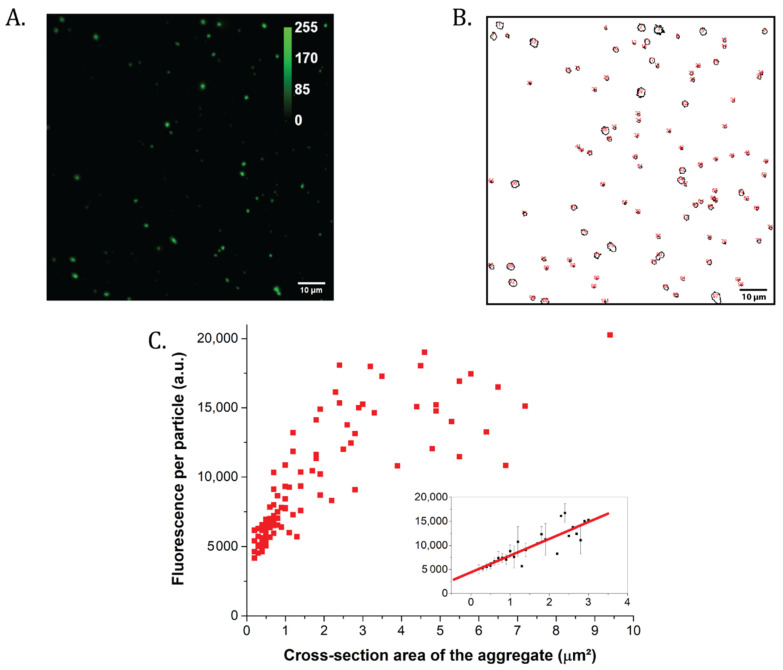
Analysis by confocal microscopy of the large LHCII particles formed below 25 μM DDM and pH 5.0. (**A**) Confocal microscopy picture of the LHCII particles precipitated at the bottom of the well. The fluorescence is presented in arbitrary units. Scalebar: 10 μm. (**B**) Analysis of the confocal picture from (A), with the cross-section of each detectable aggregate circled. (**C**) Fluorescence intensity per LHCII particles as a function of the measured cross-section area. Insert: zoom of the section between 0 and 3 μm^2^, with average particle fluorescence values and standard deviations. Red line: straight line fitting of the data; *R*^2^ = 0.8.

**Figure 5 ijms-22-02969-f005:**
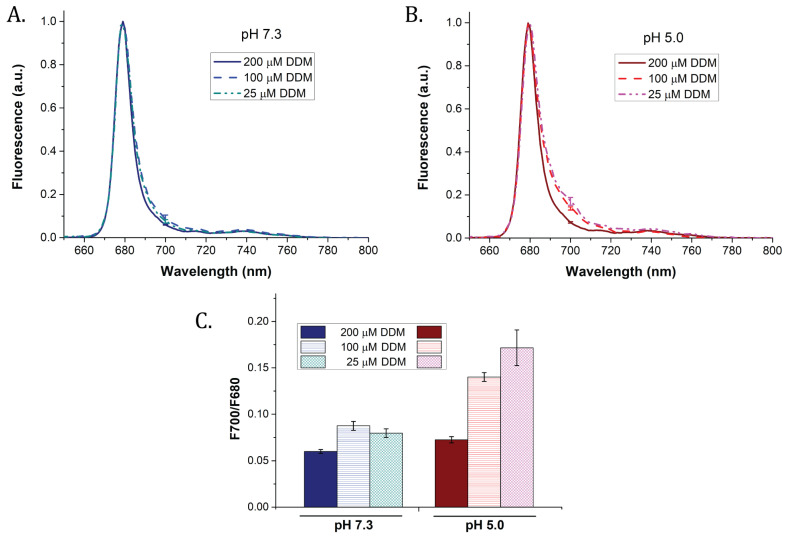
77K fluorescence emission spectra of the samples used for fluorescence quenching analyses of LHCII trimers (see Figure 1). Purified LHCII were incubated at different detergents (200, 100, or 25 μM DDM) and pH values (7.3 or 5.0) as in previous experiments. (**A**) Samples at pH 7.3; (**B**) samples at pH 5.0; (**C**) ratio of the fluorescence measured at 700 nm to the fluorescence at 680 nm for each sample. Results are the average of three replicates. Error bars: SD.

**Table 1 ijms-22-02969-t001:** Summary of the measured fluorescence fraction quenched (FQ) and calculated K_D_s in the samples measured in Figure 1.

pH	DDM Concentration	FQ	K_D_
7.3	200 µM	12.4 ± 1.2%	0.1
	100 µM	43.7 ± 0.5%	0.7
	25 µM	61.7 ± 1.7%	1.4
5.0	200 µM	20.1 ± 3.5%	0.2
	100 µM	70.9 ± 1.2%	2
	25 µM	75.6 ± 0.7%	3

**Table 2 ijms-22-02969-t002:** Properties of LHCII particles incubated at different pH and detergent concentrations on the basis of fluorescence correlation spectroscopy (FCS) measurements and fitted by Levenberg–Marquardt method (see Figure 2). Parameters were obtained assuming one single diffusional component. The calculated parameters include average diffusion time, hydro-dynamical radius of LHCII particle, total photon count rate per focal volume; N—number of LHCII particles per focal volume. Results from low-quality fittings are highlighted in gray (see the main text for more details). Error: SD.

DDM Concentration	pH	Diffusion Time (ms)	Radius (nm)	Total Count Rate (KHz)	N
200 μM	pH 7.3	1.18 ± 0.10	9.2 ± 0.8	11.48 ± 0.21	35.0 ± 0.7
	pH 5.0	1.10 ± 0.09	7.9 ± 0.0	11.67 ± 0.17	36.3 ± 3.2
100 μM	pH 7.3	1.83 ± 0.18	13.4 ± 1.4	8.16 ± 0.19	23.5 ± 3.0
	pH 5.0	3.56 ± 0.53	30.1 ± 1.5	4.11 ± 0.05	4.7 ± 1.2
25 μM	pH 7.3	1.80 ± 0.28	15.4 ± 2.4	4.69 ± 0.32	18.3 ± 2.1
	pH 5.0	7.60 ± 3.05	65.8 ± 27.0	1.56 ± 0.14	0.5 ± 0.0

## Data Availability

The data that support the findings of this study are available from the corresponding authors upon reasonable request.

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
