# Peer review of "Antenna Protein Clustering In Vitro Unveiled by Fluorescence Correlation Spectroscopy"

_ijms, 2021, doi:10.3390/ijms22062969_

Round 1
Reviewer 1 Report
In the manuscript 'Antenna protein clustering in vitro unveiled by Fluorescence Correlation Spectroscopy', Crepin et al. use a combination of FCS, confocal microscopy and fluorescence (FL) spectroscopy to investigate aggregation-induced fluorescence quenching in photosynthetic LHCII antenna complexes. While the quenching conditions in vitro are known and widely used in the field, the biological significance of aggregation for the photo-protective non-photochemical quenching (NPQ) in vivo is less clear and currently debated. In the present work, the authors succeed in relating the FL quenching to the aggregate size distribution. Performing measurements at varying pH and detergent concentration allows the authors to disentangle the effect of the two parameters on the aggregate formation. This has implications both for the common in vitro NPQ induction experiments (which use very low pH and detergent concentration) and for the potential physiological NPQ mechanisms, as the authors discuss.
The manuscript is very well written, the science is sound and thorough, the use of FCS is innovative and fits excellently the posed research question. Below I present some remarks and suggestions to the authors, in hope that these will improve the manuscript. Regardless of their response, I heartily recommend to accept the manuscript for publication in IJMS in its present form.
Major comments:
Overall, I would like to encourage the authors to discuss more specific conclusions from their work, both with respect to existing literature, and regarding the NPQ qE mechanism itself.
The in-vitro quenching induction is clear: pH drop together with lowered detergent concentration produces aggregation, which leads to quenching. How does this compare to the physiological conditions? That is, can the authors comment on the expected magnitude of the pH drop in the lumen, and the associated degree of quenching, as compared to their observation? Could this provide an estimate on the magnitude of the accompanying effects in vivo?
The absence of the red-shift at already significant quenching conditions is very intriguing. The authors mention that only after ‘significantly’ increasing the protein-to-detergent ratio, they observe the red shift (data not shown). How much is the ratio? Importantly, how does this relate to the qE observations in literature? To which extent can the mild aggregation be excluded as an explanation, based on the presence of the red shift?
Continuing with the red shift, could the authors measure the spectra of the large aggregates in the confocal microscope? It would be very interesting to see if the size correlates with the F700/F680 ratio.
Regarding the in-vitro NPQ induction, can the authors specify a correct protocol, i.e., bounds on the pH and detergent concentration to correspond to the physiological operation?
Minor comments:
In some places units missing:
-p7 l227: fluorescence intensities 9071 and 18072 counts?
-Fig. 4A scale bar in counts per?
Inset in Fig. 4C has very small labels
I think that on p14 l538 insuring should read ensuring
Author Response
Rev 1
In the manuscript 'Antenna protein clustering in vitro unveiled by Fluorescence Correlation Spectroscopy', Crepin et al. use a combination of FCS, confocal microscopy and fluorescence (FL) spectroscopy to investigate aggregation-induced fluorescence quenching in photosynthetic LHCII antenna complexes. While the quenching conditions in vitro are known and widely used in the field, the biological significance of aggregation for the photo-protective non-photochemical quenching (NPQ) in vivo is less clear and currently debated. In the present work, the authors succeed in relating the FL quenching to the aggregate size distribution. Performing measurements at varying pH and detergent concentration allows the authors to disentangle the effect of the two parameters on the aggregate formation. This has implications both for the common in vitro NPQ induction experiments (which use very low pH and detergent concentration) and for the potential physiological NPQ mechanisms, as the authors discuss.
The manuscript is very well written, the science is sound and thorough, the use of FCS is innovative and fits excellently the posed research question. Below I present some remarks and suggestions to the authors, in hope that these will improve the manuscript. Regardless of their response, I heartily recommend to accept the manuscript for publication in IJMS in its present form.
A: We would like to thank the reviewer for the very positive appreciation and feedback. We modified the manuscript according to the recommendations given. Please find our detailed answers below.
Major comments:
Overall, I would like to encourage the authors to discuss more specific conclusions from their work, both with respect to existing literature, and regarding the NPQ qE mechanism itself.
A: We added paragraphs in the introduction and discussion sections, following the reviewer’s advice, to better link our results with the existing literature on qE quenching and antenna protein aggregation.
Q1: The in-vitro quenching induction is clear: pH drop together with lowered detergent concentration produces aggregation, which leads to quenching. How does this compare to the physiological conditions? That is, can the authors comment on the expected magnitude of the pH drop in the lumen, and the associated degree of quenching, as compared to their observation? Could this provide an estimate on the magnitude of the accompanying effects in vivo?
A1: Indeed this is an interesting point to discuss as quenching extent found in small oligomers is well compatible with what can be commonly found in vivo. We have now added this point to the discussion section, as well as considerations on the expected magnitude on the lumenal pH drop in the introduction section. Moreover, Kd quenching values are now shown in the result section (see also Table 1) for a fair comparison with literature values (see Discussion).
Q2: The absence of the red-shift at already significant quenching conditions is very intriguing. The authors mention that only after ‘significantly’ increasing the protein-to-detergent ratio, they observe the red shift (data not shown). How much is the ratio? Importantly, how does this relate to the qE observations in literature? To which extent can the mild aggregation be excluded as an explanation, based on the presence of the red shift?
A2: The data concerning the red shift observed at higher protein to detergent ratio has been added in supplementary material (Figure S6) and the text modified accordingly. By using protein to detergent ratios 30 times higher than in the rest of our experiments, we reach F700/F680 fluorescence ratios of 0.48 at pH 5.0 and 100 μM DDM. This is in line with the fact that far-red fluorescence emission by LHCII in quenching conditions seems to be associated to protein aggregation (Ostroumov et al., Photosynth Res 2020) and that it seems to increase with the size of the protein aggregates (e.g. Natali et al., J Biol Chem 2016, or our own results). Because it appears when both these conditions – quenching and aggregation – are reunited, it is quite difficult to untangle its physiological relevance in relation to qE, though it has been proposed to be related to chlorophyll-chlorophyll charge transfer states involved in quenching (Chmeliov at al., Nat Plants 2016; Ostroumov et al., Photosynth Res 2020). Further studies, however, will be necessary to uncover more information about its origin and physiological relevance in vivo.
A paragraph better describing these matters has been added in the discussion section.
Q3: Continuing with the red shift, could the authors measure the spectra of the large aggregates in the confocal microscope? It would be very interesting to see if the size correlates with the F700/F680 ratio.
A3: We do have some measurements in confocal microscopy using two fluorescence channels – one from 664 to 695 nm and one from 696 to 758 nm (red-shifted fluorescence). The results have now been added in the manuscript as Figure S7. Unfortunately, these measures do not allow providing a F700/F680 ratio for the large aggregates, for two reasons. The first is technical: we had to use two different detectors of different sensitivity to get these pictures, so the emission values cannot be directly compared. The second reason is that, very interestingly, we observed intra-aggregate heterogeneity in fluorescence emission: large aggregates presented several areas with a red-shifted emission compared to the surrounding zones. These red-shifted areas could represent quenching sites, which are thought to be present in such large particles (see e.g. van Oort et al., FEBS Letters 2007; Chmeliov at al., Nat Plants 2016). However, because of the uncertainty concerning aggregate organization, number of LHCII involved, and because of the low resolution of the technique (one pixel represents a 100 nm x 100 nm area), we can only provide hypotheses; these matters will have to be investigated with better suited methods in further studies.
These points are now discussed in the manuscript in the results and discussion sections.
Q4: Regarding the in-vitro NPQ induction, can the authors specify a correct protocol, i.e., bounds on the pH and detergent concentration to correspond to the physiological operation?
A4: We added some specific considerations on this point (see discussion section) which clarify in particular the different role and drawbacks of acidification versus detergent on during in vitro quenching. A well-defined protocol, however, requires a dedicated study of protein oligomerization as a function of pH, which is currently under way.
Minor comments:
In some places units missing:
-p7 l227: fluorescence intensities 9071 and 18072 counts?
The fluorescence intensities are presented in arbitrary units. The precision has been added to the text.
-Fig. 4A scale bar in counts per?
As stated above, the unit is arbitrary. It has been added to the figure legend.
Inset in Fig. 4C has very small labels
The size of the insert labels has been increased.
I think that on p14 l538 insuring should read ensuring
It has been corrected.

Reviewer 2 Report
The manuscript by Crepin et al describes the application of the method Fluorescence correlation spectroscopy to study the clustering, oligomeric state and quenching properties of LHCII. They showed that by detergent depletion and/or acidification large LHCII clusters and aggregates can be induced, which exhibited elevated fluorescence quenching. They also demonstrated that the aggregation of LCHII by detergent removal and/or acidification caused a red shift in the fluorescence emission spectrum, although the extent of the red shift remained smaller in the in vitro aggregates as compared to previous in vivo studies.
The manuscript is well written, the addressed aims are clear, and the presented results are convincingly presented and sufficient to support the addressed hypotheses of the work. Some minor comments are provided below for further improvements.
line 45: authors should consider rephrasing ‘energetic quenching’ to ‘energy-dependent’ or high-energy state quenching’ for more accurate terminology according also to the cited literature
Figure 2: it is somewhat hard to see and distinguish the original datapoints. Measured datapoints should be presented with thicker lines and the fitted curves with thin lines. What do _1, _2, _3 stand for? replicates? This should be clarified in the figure caption.
line 133: ’quenching scaled more than linearly’ – this is not accurate term, describe the mathematically correct expression for this correlation.
line 159: ‘This result indicates that low pH deeply affects LHCII antenna protein cluster sizes.’ This is quite well known from previous literature, authors therefore should refer to the novel aspects of these findings. Relevant to this point, have the authors attempted to correlate the LHCII cluster sizes with biochemical analyses, such as size-exclusion chromatography? It is somewhat unclear how the different ‘meric’ states of LHCII was defined in this study. Some further clarifications would be necessary about the LHCII cluster sizes and about the frequently referred oligomerization status applied in the current study.
Figure 4: the panels A and B are not mentioned and explained in the main text.
The text is somewhat confusing in lines 248-253. Percentages are described in the text, while ratio values are shown in Fig. 5C, then it is mentioned that at pH=5.0 the F700/F680 is doubled. It would be better to consistently describe the values either in percentage changes overall or in ratio values.
line 254: ‘data not shown’ - this is an important aspect to show the dependency of the red shift on protein:detergent ratio, consider adding it as supplementary material/information.
line 265-266: ‘However, most protocols used to study NPQ in vitro involve detergent removal and pH drop, yielding particles of unknown size and organization’ – add relevant references here to support this statement.
Author Response
Rev 2
The manuscript by Crepin et al describes the application of the method Fluorescence correlation spectroscopy to study the clustering, oligomeric state and quenching properties of LHCII. They showed that by detergent depletion and/or acidification large LHCII clusters and aggregates can be induced, which exhibited elevated fluorescence quenching. They also demonstrated that the aggregation of LCHII by detergent removal and/or acidification caused a red shift in the fluorescence emission spectrum, although the extent of the red shift remained smaller in the in vitro aggregates as compared to previous in vivo studies.
The manuscript is well written, the addressed aims are clear, and the presented results are convincingly presented and sufficient to support the addressed hypotheses of the work. Some minor comments are provided below for further improvements.
We would like to thank the reviewer for their very positive comments and the helpful suggestions. Please find our detailed answers below.
Q1: line 45: authors should consider rephrasing ‘energetic quenching’ to ‘energy-dependent’ or high-energy state quenching’ for more accurate terminology according also to the cited literature
A1:“Energetic quenching” has been replaced with “energy-dependent quenching” as per reviewer’s advice.
Q2: Figure 2: it is somewhat hard to see and distinguish the original datapoints. Measured datapoints should be presented with thicker lines and the fitted curves with thin lines. What do _1, _2, _3 stand for? replicates? This should be clarified in the figure caption.
A2: To improve the clarity and make original datapoints more apparent, we removed the panels A et B from Figure 2 from the main text and now present the raw results and fittings of individual replicates as separate panels in Supplementary Figure 2, as well as the weighted residuals in Supplementary Figure 3.
Designations _1, _2 and _3 indeed refer to replicates; this has been added in the legends of supplementary figures 2, 3, 4 and 5.
Q3: line 133: ’quenching scaled more than linearly’ – this is not accurate term, describe the mathematically correct expression for this correlation.
A3: the sentence has been rephrased for clarification.
Q4: line 159: ‘This result indicates that low pH deeply affects LHCII antenna protein cluster sizes.’ This is quite well known from previous literature, authors therefore should refer to the novel aspects of these findings. Relevant to this point, have the authors attempted to correlate the LHCII cluster sizes with biochemical analyses, such as size-exclusion chromatography? It is somewhat unclear how the different ‘meric’ states of LHCII was defined in this study. Some further clarifications would be necessary about the LHCII cluster sizes and about the frequently referred oligomerization status applied in the current study.
A4: It is important to stress here that all the works in the field so far assumed changes in protein clustering (aggregation) during in vitro quenching. This assumption was based on the measured fluorescence changes but no direct measurement of the change could be provided before. As far as we are aware, this is the first work showing actual changes in protein sizes induced by low detergent and pH. This point has been now underlined through the paper. As this work provides the first solid base on the topic, a separate project, currently in preparation, merging biochemical analysis with FCS, will be required to explore the subject in more depth.
Q5: Figure 4: the panels A and B are not mentioned and explained in the main text.
A5: We have rephrased the corresponding results section and now explicitly mention these panels in the text.
Q6: The text is somewhat confusing in lines 248-253. Percentages are described in the text, while ratio values are shown in Fig. 5C, then it is mentioned that at pH=5.0 the F700/F680 is doubled. It would be better to consistently describe the values either in percentage changes overall or in ratio values.
A6: The paragraph has been modified as per reviewer advice, and we now consistently use ratio values when describing the red shift measured in our samples.
Q7: line 254: ‘data not shown’ - this is an important aspect to show the dependency of the red shift on protein:detergent ratio, consider adding it as supplementary material/information.
A7: As per reviewer’s advice, the data has been added in supplementary material (now Figure S6), and the corresponding text in the results part has been modified.
Q8: line 265-266: ‘However, most protocols used to study NPQ in vitro involve detergent removal and pH drop, yielding particles of unknown size and organization’ – add relevant references here to support this statement.
A8: References have been added to support the statement.
